# Association of ABCB1 Polymorphisms with Efficacy and Adverse Drug Reactions of Valproic Acid in Children with Epilepsy

**DOI:** 10.3390/ph16111536

**Published:** 2023-10-30

**Authors:** Jiahao Zhu, Jieluan Lu, Yaodong He, Xianhuan Shen, Hanbing Xia, Wenzhou Li, Jianping Zhang, Xiaomei Fan

**Affiliations:** 1Department of Clinical Pharmacology, College of Pharmacy, Jinan University, Guangzhou 511436, China; zhujiahao@stu2021.jnu.edu.cn (J.Z.); 18819748182@163.com (J.L.); dongho837@gmail.com (Y.H.); 763808557a@stu2020.jnu.edu.cn (X.S.); tzhangjp@jnu.edu.cn (J.Z.); 2Department of Pharmacy, Shenzhen Baoan Women’s and Children’s Hospital, Jinan University, Shenzhen 518102, China; xhb2588@126.com (H.X.); wenzhoulee@foxmail.com (W.L.)

**Keywords:** ABCB1, polymorphism, haplotype, childhood epilepsy, valproic acid, efficacy, adverse drug reaction

## Abstract

Genetic polymorphisms in ATP-binding cassette subfamily B member 1 (*ABCB1*, also known as *MDR1*) have been reported to be possibly associated with the regulation of response to antiseizure medications. The aim of this study was to investigate the association of *ABCB1* polymorphisms with the efficacy of and adverse drug reactions to valproic acid among Chinese children with epilepsy. A total of 170 children from southern China with epilepsy treated with valproic acid for more than one year were recruited, including 61 patients with persistent seizures and 109 patients who were seizure-free. Two single nucleotide polymorphisms of *ABCB1*, rs1128503 and rs3789243, were genotyped using the Sequenom MassArray system. The two single nucleotide polymorphisms of *ABCB1* were found to be significantly associated with treatment outcomes of valproic acid in children with epilepsy. Carriers with the TT genotype of *ABCB1* rs1128503 were more inclined to exhibit persistent seizures after treatment with valproic acid (*p* = 0.013). The CC genotype of rs3789243 was observed to be a potential protective factor for valproic acid-induced gastrointestinal adverse drug reactions (*p* = 0.018), but possibly increased the risk of valproic acid-induced cutaneous adverse drug reactions (*p* = 0.011). In contrast, the CT genotype of rs3789243 was associated with a lower risk of valproic acid-induced cutaneous adverse drug reactions (*p* = 0.011). Haplotype analysis showed that CC haplotype carriers tended to respond better to valproic acid treatment (*p* = 0.009). Additionally, no significant association was found between *ABCB1* polymorphisms and serum concentrations of valproic acid. This study revealed that the polymorphisms and haplotypes of the *ABCB1* gene might be associated with the treatment outcomes of valproic acid in Chinese children with epilepsy.

## 1. Introduction

Epilepsy is a chronic and complex neurological disorder that affects more than 70 million people worldwide [1]. The incidence of epilepsy is significantly higher in infants under 1 year of age and in people over 50 years of age [1]. In China, epidemiological surveys show that the prevalence of epilepsy is approximately 0.9–8.5 cases per 1000 people [2]. At present, antiseizure medications (ASMs) remain the most important and fundamental treatment modality for epilepsy. Although there are many ASMs available, up to 30% of patients with epilepsy fail to achieve seizure control despite receiving medication at appropriate doses and tolerability [3,4]. Drug resistance is a major challenge in the management of epilepsy, which may be related to the patient’s age, seizure type, etiology of epilepsy, and genetic factors [5,6]. On the other hand, neurologists are increasingly aware of the importance of reducing adverse drug reactions (ADRs) in the treatment of epilepsy as they affect patients’ compliance and quality of life [7]. About 80% of patients treated with ASMs were reported to have at least one side effect [8]. Yet, children may be more susceptible to ADRs than adults.

As a broad-spectrum antiepileptic drug, valproic acid (VPA) has been widely used in the treatment of many types of epilepsy and epilepsy syndromes [9]. The pharmacokinetics of VPA exhibit a non-linear profile due to its saturated plasma protein binding. The plasma protein binding of VPA is concentration dependent (74–93%). Its V_d_ is 0.13–0.19 L/kg. The plasma half-life of VPA in children is 8–13 h [10]. There are three major pathways of VPA metabolism in humans: glucuronidation, β-oxidation in the mitochondria, and the cytochrome P450 (CYP) pathway. These account for 50%, 40%, and 10% of the dose, respectively [11]. Although VPA has been widely used in various fields, its pharmacokinetic and pharmacodynamic variations are still difficult to estimate. There may be great differences in serum concentration, efficacy and safety of VPA among patients taking the same dose [12]. With the development of pharmacogenetics, genetic variation is considered to be an important factor contributing to the variability of clinical outcomes in individual patients treated with ASMs [13]. Current studies on VPA pharmacogenetics have identified numerous variations in drug transporters, drug-metabolizing enzymes and targets that are associated with the treatment outcome of VPA [14,15,16]. However, the role of genetic variations in drug transporters in VPA therapy remains controversial. It is necessary to further verify and explore these associations.

P-glycoprotein (P-gp), a typical ATP-binding cassette (ABC) transporter protein, has been found to play a prominent role in the development of multidrug resistance [17]. P-gp is encoded by the multiple drug resistance (*MDR1* or *ABCB1*) gene. In addition to the liver, kidney, small intestine, and colon, P-gp is also expressed in the endothelial cells of central nervous system capillaries that form the blood-brain barrier (BBB), which restricts the penetration of multiple drugs into the brain [6]. Tishler et al. found elevated levels of *MDR1* expression in brain specimens surgically removed from patients with intractable epilepsy. They proposed that elevated P-gp expression might limit the entry of ASMs into the brain tissue, which contributes to the refractory nature of epilepsy [18]. The increased *ABCB1* expression in specific limbic brain regions and the decreased ASM levels have also been observed in rodent models of limbic seizures [19]. The human *ABCB1* gene is located on chromosome 7 at q21.1 with 28 exons [17]. More than 50 single nucleotide polymorphisms (SNPs) have been identified in the *ABCB1* gene, the most common including C3435T (rs1045642), C1236T (rs1128503) and G2677T (rs2032582) [17,20]. Multiple studies have found that SNPs at different locations of *ABCB1* are associated with ASM responsiveness [21,22,23]. An earlier study in the European region reported that patients with drug-resistant epilepsy (DRE) were more likely to carry the CC genotype of *ABCB1* C3435T compared with patients with drug-responsive epilepsy [21]. However, two studies in Japan and China observed opposite results [22,23], which might be attributed to ethnic differences. The intro polymorphism rs3789243 and coding polymorphism G2677T/A were reported to possibly increase the risk of drug resistance in a Han Chinese epileptic population [24]. A study in a Tunisian population showed that the TT genotypes of C1236T, G2677T and C3435T were associated with DRE [25]. In addition, in our previous study, the rs3789243 variant of *ABCB1* was found to have a significant effect on the pharmacokinetic variability of VPA, which was observed by establishing a population pharmacokinetic model [26].

Pharmacogenetics is a rapidly growing field with the ultimate goal of utilizing an individual’s genetic makeup to predict drug response and efficacy, as well as potential adverse drug events. In addition, linkage disequilibrium (LD) refers to two alleles at different loci exhibiting non-random association on the same chromosome [27]. The identification of LD is also an important element in assessing the effect of gene composition on drug response. The novelty of this study lines in the fact that although several polymorphisms of *ABCB1* have been observed to be associated with DRE, few studies have focused on its association with VPA treatment outcomes. Secondly, the ADRs caused by VPA and LD between different loci were evaluated, which are rarely mentioned in previous studies. Additionally, this study was conducted in children, which is beneficial for optimizing the development of individualized treatment plans for children. Based on previous studies [23,24,25,26,28], rs1128503 (C1236T) and rs3789243 were selected as candidate SNPs of *ABCB1* to investigate their relationship with the efficacy, safety, and blood concentration of VPA in the treatment of children with epilepsy.

## 2. Results

### 2.1. Clinical Characteristics of Children with Epilepsy

In this study, a total of 170 children with epilepsy (aged 1 month–16 years) were enrolled, of whom 61 presented with persistent seizures and 109 were seizure free. The rate of poor response to VPA (35.9%) was similar to that previously reported [29]. The demographic and clinical characteristics of the patients are summarized in Table 1. The patients were predominantly male (60.0% vs. 40.0%), and the median age of initial diagnosis was 3.42 (1.00–6.44) years. Generalized onset was the dominant seizure type in both the general population (60.6%) and the groups divided by the response to VPA (59.0% and 61.5%). In terms of the subtype of seizures, the most common subtype in both groups was generalized tonic-clonic seizure, which accounted for 37.7% and 36.7% in the persistent seizure and seizure-free groups, respectively. The mean steady-state serum concentration of VPA was 62.68 ± 22.46 μg/mL. The patients in the seizure-free group received VPA monotherapy more frequently (69.7%), in contrast to the persistent seizure group, where it was more common for patients to receive polytherapy (75.4%). For polytherapy, common ASMs used in combination with VPA were levetiracetam (LEV) (26.5%) and oxcarbazepine (OXC) (20.0%). In this study, a high incidence of ASM-related ADRs (72.4%) was found in children with epilepsy. The most common ADRs induced by the ASMs in this study were gastrointestinal ADRs (gADRs) (31.8%) including appetite changes, nausea, vomiting, abdominal pain, diarrhea, constipation and dyspepsia, followed by neurological ADRs (nADRs) (30.3%) including somnolence, asthenia, dizziness, headache, irritability and sleep problems. In addition, the ADRs manifesting as weight gain and rash accounted for 24.3% and 13.6% of the total ADRs cases, respectively. There were no significant differences in age of initial diagnosis, gender, type of seizure, subtype of epilepsy, VPA concentration, and occurrence of ADRs between the persistent seizure and seizure-free groups.

Table 2 shows the relationship between the *ABCB1* polymorphisms and clinical characteristics of children with epilepsy. In the overall study population, CC (10.0%), CT (50.0%), and TT (40.0%) genotypes were detected for *ABCB1* rs1128503, and CC (41.2%), CT (47.6%), and TT (11.1%) genotypes for *ABCB1* rs3789243. The minor allele frequency (MAF) was 35% for the C allele of rs1128503 and 35% for the T allele of rs3789243, which is consistent with the information from the National Center for Biotechnology Information (NCBI) database. The frequency of genotypes in patients with different clinical characteristics displayed the same trend as the overall study population. No significant association was found between the two SNPs of the *ABCB1* gene and the clinical characteristics of the patients.

### 2.2. The Association of ABCB1 Polymorphisms with VPA Responsiveness in Children with Epilepsy

The rs1128503 and rs3789243 polymorphisms of *ABCB1* conformed with the Hardy–Weinberg equilibrium (HWE) in both persistent seizure and seizure-free groups (*p* > 0.05). The polymorphism of rs1128503 was observed to be significantly associated with the efficacy of VPA in children with epilepsy. Table 3 lists the distribution of alleles and genotypes of rs1128503 in the persistent-seizure and seizure-free groups. For rs1128503, a CC (8.2%), CT (39.3%), and TT (52.5%) pattern was observed in the persistent seizure group, and CC (11.0%), CT (56.0%), and TT (33.0%) in the seizure-free group. The frequency of the T allele was found to be higher in the persistent seizure group than that in the seizure-free group (72.1% and 61.0%, respectively). Moreover, recessive model analysis showed that the TT genotype had a higher frequency in the persistent seizure group compared with genotypes carrying at least one C allele (OR = 0.45, 95% CI = 0.24–0.85, *p* = 0.013), suggesting that the TT genotype might be associated with reduced responsiveness to VPA in children with epilepsy (Figure 1). In contrast, patients carrying the CT genotype had an increased frequency in the seizure-free group (OR = 1.96, 95% CI = 1.04–3.71, *p* = 0.037). Furthermore, no significant differences were found in the distribution of rs3789243 genotypes between the persistent seizure and seizure-free groups (Appendix A).

The results of the LD and haplotype analysis showed that the two SNPs of *ABCB1* exhibited a weak LD, with a low D’ value of 0.086 and an R^2^ value of 0.002 between the two alleles. This indicated that the two alleles were more likely to be inherited independently. The haplotype frequencies composed of rs1128503 and rs3789243 were calculated using SHEsis and analyzed for correlation with VPA treatment response (Table 4). Among the detected haplotypes, TC and CC patterns showed associations with the response to VPA treatment. The CC haplotype was significantly more frequent in the seizure-free group compared with other haplotypes (OR = 0.48, 95% CI = 0.27–0.84, *p* = 0.009), while patients with the TC haplotype appeared to be more inclined to exhibit poor response to VPA (OR = 1.58, 95% CI = 1.00–2.47, *p* = 0.047).

### 2.3. The Association of ABCB1 Polymorphisms with VPA-Induced ADRs in Children with Epilepsy

The relevance of *ABCB1* polymorphisms to VPA-induced ADRs in children with epilepsy was further explored. The polymorphism of *ABCB1* rs3789243 was found to be associated to some extent with VPA-induced gADRs and cutaneous ADRs (cADRs). The distribution of alleles and genotypes of rs3789243 in the patients with and without VPA-induced gADRs is shown in Table 5. Specifically, the frequency of the T allele of rs3789243 was higher in the patients with VPA-induced gADRs compared with the C allele (42.9% vs. 31.1%, *p* = 0.033). The dominant model analysis revealed frequencies of 28.6% and 47.4% for the CC genotype in patients with and without gADRs, respectively, compared with 71.4% and 52.6% for the CT + TT genotype. The patients who were carriers of the CC genotype had a lower risk of gADRs during VPA treatment compared with those with the CT + TT genotype (OR = 0.44, 95% CI = 0.22–0.88, *p* = 0.018) (Figure 2). Additionally, similar results were observed in the log-additive model.

As described in Table 6, the polymorphism of *ABCB1* rs3789243 was also significantly associated with VPA-induced cADRs. There was a significant difference in the frequency of CC, CT and TT genotypes of rs3789243 among the patients with and without cADRs (*p* = 0.027). The patients who were carriers of the CC genotype exhibited a higher risk of cADRs during VPA treatment compared with those with the CT + TT genotype (OR = 3.57, 95% CI = 1.29–9.93, *p* = 0.011) (Figure 3), whereas the CT genotype was associated with a lower risk of cADRs (OR = 3.90, 95% CI = 1.24–12.30, *p* = 0.011). Log-additive analysis also showed an increased frequency of the CC genotype compared with the TT genotype in the patients with cADRs. These results suggest that the CC genotype of rs3789243 might be a risk factor for cADRs during treatment with VPA, while the CT genotype might reduce this risk. Furthermore, there were no statistically significant associations found between the rs1128503 polymorphism and VPA-induced gADRs and cADRs (Appendix A). Additionally, no significant association was found between the two polymorphisms and nADRs induced by VPA (Appendix A). Similarly, no significant association was observed between *ABCB1* haplotypes and VPA-induced ADRs.

### 2.4. The Association between ABCB1 Polymorphisms and Serum Concentrations of VPA

The comparison of serum concentration and adjusted concentration (AC) of VPA for the different genotypes of *ABCB1* is tabulated in Table 7. Most of the patients (65.6%) enrolled in this study had serum concentrations of VPA within the reference range of 50 to 100 μg/mL [10]. Since the serum concentrations of VPA were non-normally distributed after adjustment for dosage and body weight, the Kruskal–Wallis *H* test was used to compare the differences in the AC of VPA across genotypes. Overall, no significant differences in VPA concentration and AC were observed with respect to the genotypes of *ABCB1* rs1128503 and rs3789243 among all the participants.

## 3. Discussion

Epilepsy is a brain condition with a complex pathogenesis of central nervous system dysfunction. Clinicians are increasingly aware of the significance of improving patients’ quality of life in the management of epilepsy. In general, the principal evaluation indicators related to patients’ quality of life are seizure control and adverse drug reactions [4]. Drug resistance in epilepsy has been a persistent clinical challenge. Some hypotheses such as the transporter hypothesis try to explain the occurrence of this phenomenon [5,18]. P-gp plays a crucial role in the absorption and transport of drugs. Polymorphisms in the *ABCB1* gene encoding P-gp have been demonstrated to influence the outcome of epilepsy treatment [18,30]. The *ABCB1* gene was found to be overexpressed in blood vessels, astrocytes and neurons in human drug-resistant epileptic brains, leading to a decrease in ASM concentrations in the brain [30]. Several studies have shown that P-gp can mediate the transport of some ASMs such as phenytoin [31], phenobarbital, and lamotrigine [32] across the BBB, suggesting that *ABCB1* polymorphisms may play a critical role in the variability in responsiveness to ASMs. However, the relationship between *ABCB1* gene expression and the transport of VPA is controversial [33,34].

In the present study, a moderate incidence (35.9%) of poor response to VPA and a high incidence (72.4%) of ADRs were found. There were no significant differences in the clinical characteristics of the patients in the persistent seizure and seizure-free groups. The selection of the two polymorphisms was based on previous reports suggesting their association with ASM responsiveness [23,24,25,26]. The polymorphism of *ABCB1* rs1128503 as well as a haplotype consisting of the two SNPs were found to be associated with the efficacy of VPA. Although the expression of *ABCB1* in the BBB is correlated with the transport of some ASMs, the association of the rs1128503 polymorphism with ASM concentrations and responsiveness remains controversial. In a prospective study in Croatia, the *ABCB1* rs1128503 polymorphism was found to significantly affect lamotrigine concentration [35]. The concentration of lamotrigine was significantly higher in epileptic patients with CC genotype compared with those with CT and TT genotypes. Puranik et al. observed significantly higher carbamazepine clearance in patients with epilepsy carrying at least one T allele of rs1128503 compared with those with the CC genotype in African-Americans [36]. Chouchi et al. identified the TT genotype and T allele of rs1128503 in a Tunisian population with epilepsy as a potential risk for increased development of DRE [25]. Additionally, it was observed that those carrying the C allele of rs1128503 in Jordanian women with epilepsy tended to be more unresponsive to ASMs than those with the T allele [37]. These results suggest that the polymorphism of *ABCB1* rs1128503 may play a role in ASM responsiveness. However, there are also studies that did not confirm the association between the rs1128503 polymorphism and the response to ASMs [38,39].

In contrast to previous reports, in the present study, the T allele and TT genotype of rs1128503 were found to be more frequent in the persistent seizure group. This result suggests that the T allele and TT genotype of rs1128503 might be potential risk factors for poor response to VPA, which is analogous to a report from Tunisia [25]. *ABCB1* rs1128503 (C1236T), a synonymous SNP, is located in exon 12 of the *ABCB1* gene [40]. It was shown that polymorphisms of *ABCB1* in C1236T, G2677T and C3435T and haplotypes composed of these three SNPs significantly affected the activity of P-gp in vitro and might reduce the efflux transport function of P-gp [20,41]. One theory suggested that the change in P-gp function is caused by a “silent” polymorphism in *ABCB1*, possibly due to the conversion of frequent codons to rare codons, which affects the time of co-translational folding [20]. Since P-gp is expressed in the capillary endothelial cells of the central nervous system [40], it is reasonable to hypothesize that the polymorphism in C1236T may have an impact on responsiveness to ASMs. There are some discrepancies between our results and some of the studies mentioned above. A number of factors contribute to the nonuniformity across studies, including limited sample sizes and racial differences. Individual studies have different definitions of the therapeutic effect of epilepsy, which leads to errors in the comparison of the results of different studies. In contrast to previous studies, we chose VPA as the subject of the ASMs studied. Additionally, the association between the polymorphisms of *ABCB1* rs1128503 and responsiveness to ASMs might not be significant in Asian populations [38,42]. Considering that the influence of genetic factors is complicated, the polymorphisms of rs1128503 might also interact with other SNPs to affect the treatment of ASMs [25].

Haplotype analysis revealed a weak LD between the rs1128503 (C1236T) and rs3789243 polymorphisms of the *ABCB1* gene. The CC haplotype was significantly more prevalent in the seizure-free group compared with other haplotypes. This suggests that genetic factors may play a role in drug response through combinations of polymorphisms rather than just individual SNPs. Most of the previous studies on the haplotypes of the *ABCB1* gene focused on the C1236T, G2677T and C3435T loci. Zimprich et al. reported that the CGC haplotype (C1236T, G2677T, C3435T) significantly increased the risk of treatment failure in temporal lobe epilepsy [43]. In contrast, CTT, TTC, TGC and TTT haplotypes were found to be significantly associated with ASM resistance in a Tunisian population [25]. However, no statistical association between the three haplotype models (C1236T-G2677T-C3435T) and ASM response was detected in a meta-analysis of *ABCB1* haplotypes [42]. Our results suggest that CC haplotype carriers tend to respond better to treatment of VPA. Considering these results, detection of *ABCB1* haplotypes might provide additional information on new phenotype–genotype correlations to assist clinicians in the rational use of ASMs. It should be noted that the results of LD in this study lack robustness, with the result that the haplotype studies provide limited information. Further validation of these associations in larger samples should be considered.

The occurrence of adverse events with ASMs is known to be an important aspect in assessing the quality of life of patients [44]. In the present study, VPA-related ADRs were divided into subgroups to evaluate the impact of variants in the *ABCB1* gene in specific ADRs. The polymorphisms in *ABCB1* rs3789243 were found to be associated with VPA-related gADRs and cADRs. The C allele and CC genotype of rs3789243 were observed to be potential protective factors for VPA-induced gADRs, but possibly increased the risk of VPA-induced cADRs. In contrast, the CT genotype of rs3789243 might decrease the risk of VPA-induced cADRs. This finding that there is an association between a polymorphism of *ABCB1* rs3789243 and VPA-induced ADRs is novel, although a significant association between the C allele of rs3789243 and multidrug-resistant epilepsy has been reported in a Han Chinese population [24]. The G allele and the haplotype containing the G allele of this intronic variant have been reported to be significantly associated with ulcerative colitis in Caucasian populations [45]. The ABCB1 protein is highly expressed in apical cells of the small intestine and the colon, which is critical for defense against xenobiotics including bacterial products [45,46]. *Mdr1a* knockout mice were observed to develop severe intestinal inflammation under specific pathogen-free conditions [47]. These findings indicated that variants in the *ABCB1* gene might play a potential role in the development of gastrointestinal diseases. Additionally, there are a few reports on the association of *ABCB1* polymorphisms with cutaneous diseases. The polymorphism in *ABCB1* exon 21 was found to be a genetic factor associated with cutaneous T-cell lymphoma [48]. Considering the small number of patients with cADRs in the present study, further validation of this association in other cohorts is warranted. Furthermore, we found no significant association between *ABCB1* polymorphisms and serum concentrations and AC of VPA. This finding might be attributed to several factors. Firstly, the sample size in this study was not large enough to observe a significant association. Secondly, the pharmacokinetics of VPA are also affected by some metabolic enzymes such as UGT1A6 and CYP2C9 [49,50]. It has been shown that there is an increase in protein expression of metabolizing enzymes such as UGT1A4 and CYP2C9 in BBB endothelial cells of epileptic patients [51,52]. Thus, the effect of the transporter protein ABCB1 on VPA concentration could also be influenced by changes in metabolic enzyme expression. It can be considered that the metabolic and antiseizure effects of VPA involve complex induction processes.

The small sample size in this study limited the determination of a significant association between *ABCB1* polymorphisms and treatment outcomes of VPA. Although adverse effects of VPA were rigorously assessed according to inclusion criteria, subjective reporting by patients’ parents might still lead to overestimation of the incidence of ADRs. Additionally, the determination of ADRs for VPA was interfered with to some extent by other ASMs when the patients were treated with polytherapy. Furthermore, the molecular mechanisms regarding the impact of *ABCB1* polymorphisms on VPA treatment outcomes remain to be elucidated.

## 4. Materials and Methods

### 4.1. Subjects

A total of 170 children with epilepsy in southern China were recruited in this prospective study, which was conducted at Shenzhen Baoan Women’s and Children’s Hospital from October 2016 to December 2022. Seizures were diagnosed by two experienced neurologists and classified according to guidelines of the International League against Epilepsy (ILAE) Commission [53]. This study was approved by the Shenzhen Baoan Women’s and Children’s Hospital Ethics Committee. Informed consent was obtained from all participants or their guardians. Children with epilepsy aged 1 month to 16 years at initial diagnosis who had been treated with VPA for at least 12 months were enrolled. Appropriate medication regimens were adopted, taking into account seizures, medication factors, and patient-specific factors. The exclusion criteria were as follow: (1) patients diagnosed with epilepsy within one month of birth, (2) patients treated with a combination of medications that affect the metabolism of VPA, (3) history or evidence of impairment of hepatic or renal function, (4) Dravet syndrome, (5) significant structural lesions in the brain, and (6) indeterminate diagnosis and incomplete clinical data. Demographic and clinical characteristics were collected through the hospital information system and a questionnaire, which included gender, age, age of initial diagnosis, body weight, types and etiology of seizures, treatment regimen and ADRs. Given that the patients in this study were children, the questionnaires were generally filled out by their relatives or physicians depending on the actual condition of the patients. Once patients met the requirements, changes in medication regimens and demographic characteristics were recorded and the efficacy and ADRs of VPA were evaluated every three months until the end of the study. Patients were followed up for a period of one year after a stabilizing dose of VPA was reached.

### 4.2. Determination of VPA Serum Concentration

Before the collection of blood samples, the blood concentration of VPA in all patients had reached a stable state (the time required to reach a stable state was 2–3 days). Venous blood (2 mL) was collected on an empty stomach before the medication the following morning. Free serum concentration of VPA was determined by homogeneous enzyme immunoassay using a HITACHI biochemical analyzer (7180 HITACHI, Tokyo, Japan). The linear range was 0.50–160 μg/mL (r ≥ 0.990). The inter-batch variations were ≤10.0% and intra-batch variations were ≤8.0%. To eliminate errors introduced by body weight and dosages, the following equation was used to calculate the adjusted concentration (AC) of VPA: AC = serum VPA concentration (μg/mL)/[daily VPA dose (mg)/weight (kg)] [54].

### 4.3. Drug Response and ADRs Evaluation

According to the definition established by the ILAE [55], patients were classified into persistent seizure and seizure-free groups. Patients were classified as persistent seizure when VPA monotherapy or therapy combined with other ASMs for at least 12 months failed to achieve sustained seizure freedom at the maximal tolerated dose. Patients were considered seizure-free if they did not experience any type of seizure during VPA treatment for at least 1 year [56,57]. The evaluation of VPA responsiveness was performed after a stable dose of VPA had been reached [58]. In addition, the occurrence of ADRs was determined by reviewing the treatment logs of neurologists, questionnaires and spontaneous records from patients’ relatives. According to the occurrence of VPA-related ADRs, the subjects were divided into groups with and without ADRs. Further, the ADRs were divided into neurological ADRs (nADRs); including somnolence, asthenia, dizziness, headache, irritability, sleep problems, and disturbance in attention; gastrointestinal ADRs (gADRs), including appetite changes, nausea, vomiting, abdominal pain, diarrhea, constipation and dyspepsia; cutaneous ADRs (cADRs), including rash, alopecia and hypersensitivity; and weight gain subgroups based on the clinical manifestations of patients. ADR reports were analyzed using the WHO-UMC causality assessment scale [59,60]. ADRs that were identified as certain, probable/likely and possible were considered subjects for subsequent studies of the association between genetic factors and VPA-induced ADRs.

### 4.4. SNP Selection and Genotyping

The two SNPs, rs1128503 and rs3789243, were selected as candidate SNPs for *ABCB1* based on the following criteria: The SNPs selected have been reported to be potentially associated with responsiveness to VPA or ASMs in previous studies [23,24,25,26]. An MAF > 0.05 was obtained from the NCBI database (https://www.ncbi.nlm.nih.gov/snp) (accessed on 29 August 2023). The frequency of genotype distribution in the population conforms to the HWE. Peripheral venous blood (1.5 mL) was collected from 170 children with epilepsy. Genomic DNA was extracted from whole blood samples of the patients using a DNA extraction protocol [61]. Polymerase chain reaction (PCR) was performed to amplify DNA samples, employing a system and conditions described in detail in our previous study [56,60]. For each PCR reaction, 10 ng of template DNA and 0.5 µM of primers were used. PCR was than performed, including initial denaturation at 95 °C for 2 min, followed by 45 amplification cycles at 95 °C for 30 s, annealing at 56 °C for 30 s, extension at 72 °C for 1 min, and final extension at 72 °C for 5 min. The PCR products were purified using shrimp alkaline phosphatase according to the manufacturer’s instructions. The cycle sequencing was than carried out using Complete iPLEX Gold Genotyping Reagents (Agena Bioscience^TM^, San Diego, CA, USA). The two SNPs of *ABCB1* gene were genotyped by Sequenom MassArray system (Agena Bioscience, San Diego, CA, USA) and iPLEX Gold Assay. MassArray system consists of the MassArray Analyzer mass spectrometer and integrated data analysis software. The genotyping primers of rs1128503 and rs3789243 used in this study were as follows: forward primer sequence: 5′-ACGTTGGATGAGCCACTGTTTCCAACCAGG-3′ and reverse primer sequence: 5′-ACGTTGGATGTTTCTCACTCGTCCTGGTAG-3′ for rs1128503; forward primer sequence: 5′-ACGTTGGATGATAAGCCCAAGATCCTGTCC-3′ and reverse primer sequence: 5′-ACGTTGGATGTCTCTGACTGCTTCAGTTCC-3′ for rs3789243.

### 4.5. Statistical Analysis

Clinical characteristics and genotype data were analyzed using Statistical Package for Social Science (SPSS) software (V26.0). The data were tested for normality before statistical analysis. The independent sample *t*-test or Mann–Whitney *U* test was performed to compare the difference between the continuous variables in persistent seizure and seizure-free groups. The Chi-square test was used to compare differences between groups in categorical variables such as gender. The SNPStats online tool (https://www.snpstats.net/start.htm) (accessed on 29 August 2023) was used to assess the deviation from the HWE of selected SNPs and analyze the distribution of alleles and genotypes in different groups. The association between genotypes and efficacy and ADR subtypes of VPA were analyzed using the chi-square test, with the wild-type or genotypes containing wild-type as the reference group. In addition, LD detection and haplotype construction were carried out using the SHEsis online platform [62]. The differences in steady-state serum concentrations of VPA between different genotypes was compared by Analysis of Variance (ANOVA) with Bonferroni’s post hoc comparison (for normal distribution) and Kruskal–Wallis *H* test (for non-normal distributions). *p*-value < 0.05 (two-sides) was considered statistically significant.

## 5. Conclusions

This study indicated that the polymorphism of *ABCB1*, rs1128503, and a haplotype consisting of *ABCB1* rs1128503 and rs3789243 were associated with VPA responsiveness. The rs3789243 polymorphism of *ABCB1* was associated with VPA-induced gADRs and cADRs. No significant association was found between the two SNPs and serum concentrations of VPA. These results explain, to some extent, the source of individual variability in VPA response.

## Figures and Tables

**Figure 1 pharmaceuticals-16-01536-f001:**
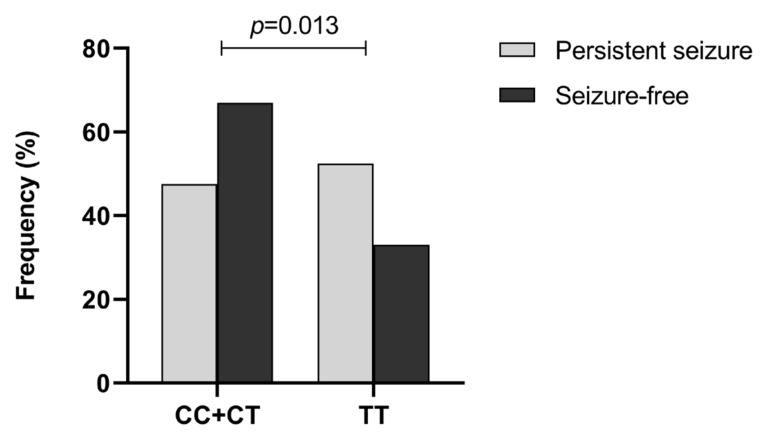
Distribution of rs1128503 genotype in the persistent seizure and seizure-free groups. Statistical method: chi-square test.

**Figure 2 pharmaceuticals-16-01536-f002:**
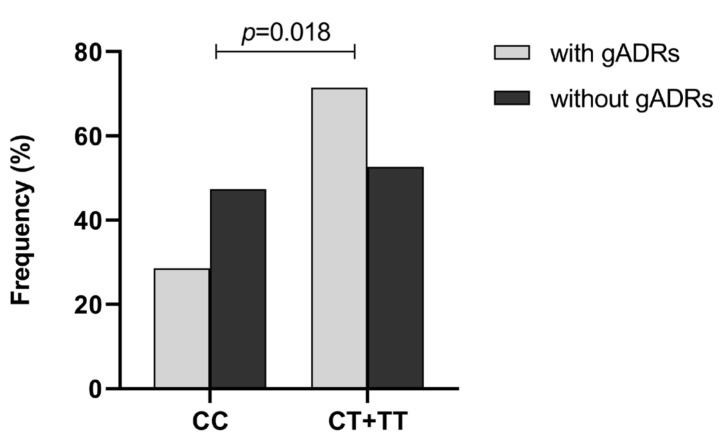
Distribution of rs3789243 genotype in children with epilepsy with or without VPA-induced gADRs. Statistical method: chi-square test.

**Figure 3 pharmaceuticals-16-01536-f003:**
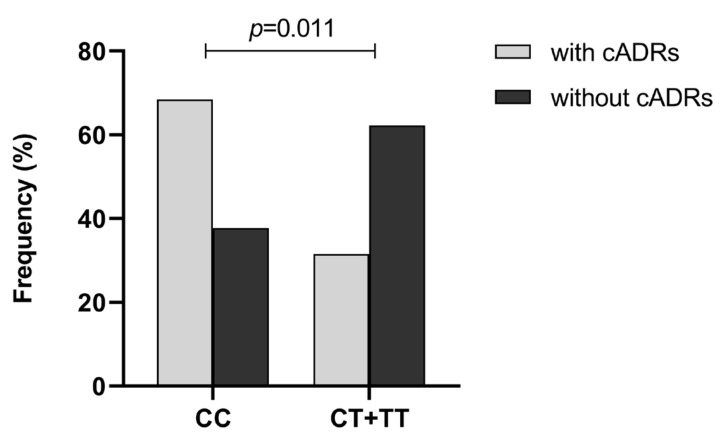
Distribution of rs3789243 genotypes in children with epilepsy with or without VPA-induced cADRs. Statistical method: chi-square test.

**Table 1 pharmaceuticals-16-01536-t001:** Characteristics of the children with epilepsy treated with VPA.

Demographic Data	Total (*n* = 170)	Persistent Seizure (*n* = 61)	Seizure Free (*n* = 109)	*p*-Value ^3^
Age of initial diagnosis				
1 month–2 years	68 (40.0%)	27 (44.3%)	41 (37.6%)	0.40
2–16 years	102 (60.0%)	34 (55.7%)	68 (62.4%)	
Median (quartile 1–3)	3.42 (1.00–6.44)	2.50 (1.00–5.17)	3.58 (1.08–7.04)	0.11
Gender				
Male	102 (60.0%)	38 (62.3%)	64 (58.7%)	0.65
Female	68 (40.0%)	23 (37.7%)	45 (41.3%)	
Seizure type				
Generalized onset	103 (60.6%)	36 (59.0%)	67 (61.5%)	0.89
Focal onset	52 (30.6%)	20 (32.8%)	32 (29.4%)	
Unknown onset	15 (8.8%)	5 (8.2%)	10 (9.2%)	
Subtype of epilepsy				
Generalized tonic-clonic seizure	63 (37.1%)	23 (37.7%)	40 (36.7%)	0.90
Tonic seizure	5 (2.9%)	0 (0.0%)	5 (4.6%)	0.16
Clonic seizure	9 (5.3%)	1 (1.6%)	8 (7.3%)	0.22
Myoclonic seizure	3 (1.8%)	2 (3.3%)	1 (0.9%)	0.61
Childhood absence epilepsy	12 (7.1%)	4 (6.6%)	8 (7.3%)	1.00
Focal seizures	26 (15.3%)	9 (14.8%)	17 (15.6%)	0.88
Benign epilepsy in childhood with centrotemporal spikes	17 (10.0%)	5 (8.2%)	12 (11.0%)	0.56
Focal to bilateral tonic-clonic seizure	4 (2.4%)	2 (3.3%)	2 (1.8%)	0.95
Others	31 (18.2%)	15 (24.6%)	16 (14.7%)	0.11
Serum concentration of VPA (µg/mL)				
Mean ± SD	62.68 ± 22.46	60.79 ± 23.00	63.69 ± 22.15	0.22
ASM therapy				
VPA monotherapy	91 (53.5%)	15 (24.6%)	76 (69.7%)	<0.01 *
Polytherapy	79 (46.5%) ^1^	46 (75.4%)	33 (30.3%)	
Levetiracetam	45 (26.5%)	30 (49.2%)	15 (13.8%)	
Oxcarbazepine	34 (20.0%)	18 (29.5%)	16 (14.7%)	
Topiramate	10 (5.9%)	10 (16.4%)	0 (0.0%)	
Lamotrigine	7 (4.1%)	5 (8.2%)	2 (1.8%)	
Clonazepam	3 (1.8%)	0 (0.0%)	3 (2.8%)	
Carbamazepine	2 (1.2%)	2 (3.3%)	0 (0.0%)	
Adverse drug reactions				
With ADRs	123 (72.4%)	42 (68.9%)	81 (74.3%)	0.45
Without ADRs	47 (27.6%)	19 (31.1%)	28 (25.7%)	
Gastrointestinal ADRs	63 (31.8%) ^2^			
Neurological ADRs	60 (30.3%)			
Weight gain	48 (24.3%)			
Rash	27 (13.6%)			

^1^ The sum of the number of patients in each group treated with ASMs was greater than the sample number (*n* = 170) due to the presence of drug combinations. ^2^ Considering that some patients experienced multiple ADRs, the total number of ADRs (198) was larger than the sample size. ^3^
*p*-value was obtained from persistent seizure group (*n* = 61) versus seizure-free group (*n* = 109). * *p* < 0.05.

**Table 2 pharmaceuticals-16-01536-t002:** Associations between *ABCB1* polymorphisms and clinical characteristics of the patients.

Demographic Data	*ABCB1* rs1128503			*p*-Value	*ABCB1* rs3789243			*p*-Value
	CC	CT	TT		CC	CT	TT	
Age of initial diagnosis								
1 month–2 years	8 (11.8%)	30 (44.1%)	30 (44.1%)	0.445	30 (44.1%)	32 (47.1%)	6 (8.8%)	0.668
2–16 years	9 (8.8%)	55 (53.9%)	38 (37.3%)		40 (39.2%)	49 (48.0%)	13 (12.7%)	
Gender								
Male	11 (10.8%)	46 (45.1%)	45 (44.1%)	0.292	42 (41.2%)	50 (49.0%)	10 (9.8%)	0.767
Female	6 (8.8%)	39 (57.4%)	23 (33.8%)		28 (41.2%)	31 (45.6%)	9 (13.2%)	
Seizure type								
Generalized onset	11 (10.7%)	47 (45.6%)	45 (43.7%)	0.298	45 (43.7%)	45 (43.7%)	13 (12.6%)	0.363
Focal onset	3 (5.8%)	30 (57.7%)	19 (36.5%)		18 (34.6%)	29 (55.8%)	5 (9.6%)	
ASM therapy								
VPA monotherapy	11 (12.1%)	45 (49.5%)	35 (38.5%)	0.612	37 (40.7%)	43 (47.3%)	11 (12.1%)	0.921
Polytherapy	6 (7.6%)	40 (50.6%)	33 (41.8%)		33 (41.8%)	38 (48.1%)	8 (10.1%)	
ADRs								
With ADRs	13 (10.6%)	64 (52.0%)	46 (37.4%)	0.534	52 (42.3%)	57 (46.3%)	14 (11.4%)	0.858
Without ADRs	4 (8.5%)	21 (44.7%)	22 (46.8%)		18 (38.3%)	24 (51.1%)	5 (10.6%)	

**Table 3 pharmaceuticals-16-01536-t003:** Comparison of *ABCB1* rs1128503 (C1236T) genotypes distribution in the persistent seizure and seizure-free groups.

Genetic Model	Genotype	Persistent Seizure (*n* = 61)	Seizure-Free (*n* = 109)	OR (95% CI)	*p*-Value
Allele contrast	C vs. T	34 (27.9%)	85 (39.0%)	1.00	0.039 *
		88 (72.1%)	133 (61.0%)	0.60 (0.37–0.98)	
Codominant	CC vs. CT vs. TT	5 (8.2%)	12 (11.0%)	1.00	0.047 *
		24 (39.3%)	61 (56.0%)	1.06 (0.34–3.33)	
		32 (52.5%)	36 (33.0%)	0.47 (0.15–1.48)	
Dominant	CC vs. CT + TT	5 (8.2%)	12 (11.0%)	1.00	0.550
		56 (91.8%)	97 (89.0%)	0.72 (0.24–2.16)	
Recessive	CC + CT vs. TT	29 (47.5%)	73 (67.0%)	1.00	0.013 *
		32 (52.5%)	36 (33.0%)	0.45 (0.24–0.85)	
Overdominant	CC + TT vs. CT	37 (60.7%)	48 (44.0%)	1.00	0.037 *
		24 (39.3%)	61 (56.0%)	1.96 (1.04–3.71)	
Log-additive	CC vs. TT	5 (8.2%)	12 (11.0%)	1.00	0.028 *
		32 (52.5%)	36 (33.0%)	0.57 (0.34–0.95)	

* *p* < 0.05.

**Table 4 pharmaceuticals-16-01536-t004:** Comparison of *ABCB1* haplotypes distribution in the persistent seizure and seizure-free group.

Gene	Haplotypes	Persistent Seizure (Freq)	Seizure-Free (Freq)	OR (95% CI)	*p*-Value
*ABCB1*	TT ^1^	0.244	0.243	1.00 (0.60–1.68)	0.989
	TC	0.478	0.367	1.58 (1.00–2.47)	0.047 *
	CT	0.117	0.101	1.18 (0.58–2.39)	0.648
	CC	0.162	0.289	0.48 (0.27–0.84)	0.009 *

^1^ The order of polymorphisms is rs1128503, rs3789243. All those with frequency < 0.03 are ignored in the analysis. * *p* < 0.05.

**Table 5 pharmaceuticals-16-01536-t005:** Comparison of *ABCB1* rs3789243 genotypes distribution in children with epilepsy with or without VPA-induced gADRs.

Genetic Model	Genotype	Patients with gADRs (*n* = 56)	Patients without gADRs (*n* = 114)	OR (95% CI)	*p*-Value
Allele contrast	C vs. T	64 (57.1%)	157 (68.9%)	1.00	0.033 *
		48 (42.9%)	71 (31.1%)	0.60 (0.38–0.96)	
Codominant	CC vs. CT vs. TT	16 (28.6%)	54 (47.4%)	1.00	0.059
		32 (57.1%)	49 (43.0%)	0.45 (0.22–0.93)	
		8 (14.3%)	11 (9.7%)	0.41 (0.14–1.19)	
Dominant	CC vs. CT + TT	16 (28.6%)	54 (47.4%)	1.00	0.018 *
		40 (71.4%)	60 (52.6%)	0.44 (0.22–0.88)	
Recessive	CC + CT vs. TT	48 (85.7%)	103 (90.3%)	1.00	0.380
		8 (14.3%)	11 (9.7%)	0.64 (0.24–1.70)	
Overdominant	CC + TT vs. CT	24 (42.9%)	65 (57.0%)	1.00	0.082
		32 (57.1%)	49 (43.0%)	0.57 (0.30–1.08)	
Log-additive	CC vs. TT	16 (28.6%)	54 (47.4%)	1.00	0.030 *
		8 (14.3%)	11 (9.7%)	0.58 (0.36–0.95)	

* *p* < 0.05.

**Table 6 pharmaceuticals-16-01536-t006:** Comparison of *ABCB1* rs3789243 genotypes distribution in children with epilepsy with or without VPA-induced cADRs.

Genetic Model	Genotype	Patients with cADRs (*n* = 19)	Patients without cADRs (*n* = 151)	OR (95% CI)	*p*-Value
Allele contrast	C vs. T	30 (78.9%)	191 (63.2%)	1.00	0.056
		8 (21.1%)	111 (36.8%)	2.18 (0.97–4.92)	
Codominant	CC vs. CT vs. TT	13 (68.4%)	57 (37.8%)	1.00	0.027 *
		4 (21.1%)	77 (51.0%)	4.39 (1.36–14.17)	
		2 (10.5%)	17 (11.3%)	1.94 (0.40–9.45)	
Dominant	CC vs. CT + TT	13 (68.4%)	57 (37.8%)	1.00	0.011 *
		6 (31.6%)	94 (62.2%)	3.57 (1.29–9.93)	
Recessive	CC + CT vs. TT	17 (89.5%)	134 (88.7%)	1.00	0.920
		2 (10.5%)	17 (11.3%)	1.08 (0.23–5.08)	
Overdominant	CC + TT vs. CT	15 (78.9%)	74 (49.0%)	1.00	0.011 *
		4 (21.1%)	77 (51.0%)	3.90 (1.24–12.30)	
Log-additive	CC vs. TT	13 (68.4%)	57 (37.8%)	1.00	0.042 *
		2 (10.5%)	17 (11.3%)	2.27 (0.98–5.26)	

* *p* < 0.05.

**Table 7 pharmaceuticals-16-01536-t007:** Comparison of *ABCB1* gene polymorphisms with VPA concentration and adjusted VPA concentration.

Gene	SNP	Genotype	VPA Concentration (μg/mL)	Adjusted VPA Concentration (μg/mL)/(mg/kg)
*ABCB1*	rs1128503	CC (10.0%)	67.19 ± 19.51 ^1^	2.55 (2.16–3.90) ^3^
		CT (50.0%)	62.79 ± 22.81	2.56 (1.99–3.58)
		TT (40.0%)	61.60 ± 22.60	2.57 (1.99–3.48)
		*p*-value	0.417 ^2^	0.642 ^4^
	rs3789243	CC (41.2%)	60.98 ± 22.09	2.53 (1.89–3.44)
		CT (47.6%)	64.14 ± 23.35	2.61 (1.99–3.67)
		TT (11.1%)	62.95 ± 20.48	2.59 (2.24–3.31)
		*p*-value	0.422 ^2^	0.518 ^4^

^1^ Mean ± SD; ^2^ One-way ANOVA; ^3^ Median (quartile 1–3); ^4^ Kruskal–Wallis *H* test.

## Data Availability

The data presented in this study are available upon request from the corresponding author.

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
