# Peer review of "Association of ABCB1 Polymorphisms with Efficacy and Adverse Drug Reactions of Valproic Acid in Children with Epilepsy"

_pharmaceuticals, 2023, doi:10.3390/ph16111536_

Round 1

Reviewer 1 Report

Comments and Suggestions for Authors The topic of the article is very interesting because there is not much information related to the causes of drug-resistant epilepsy or the response to antiepileptic drugs. I think that the authors could analyze the association of the presence of SNPs with some clinical characteristics of the patients.

Author Response

Dear reviewer,

Thank you very much for your careful review and constructive suggestions to our manuscript. We have addressed the comments and highlighted all the corrections in red in the revised manuscript. Response to your comments and suggestions are listed below.

Comments 1: The topic of the article is very interesting because there is not much information related to the causes of drug-resistant epilepsy or the response to antiepileptic drugs. I think that the authors could analyze the association of the presence of SNPs with some clinical characteristics of the patients.

Response 1: Thank you very much for your comments and we agree with you. While most studies have focused primarily on the influence of genetic factors on drug response, studies of the relationship between the presence of SNPs and the clinical characteristics of patients are beneficial in presenting a complete picture of the article. For example, these studies can help us understand whether genetic polymorphisms influence the type of epilepsy. Therefore, we have supplemented our study of the association between the presence of the ABCB1 SNPs and the clinical characteristics of the patients (page 3-4 in the revised manuscript). Unfortunately, we did not find a significant association between the two SNPs of ABCB1 and the clinical characteristics of the patients. Further studies could be considered after expanding the sample.

Reviewer 2 Report

Comments and Suggestions for Authors

Manuscript ID: pharmaceuticals-2622174

Title “Association of ABCB1 Polymorphism with Therapeutic Response to Valproic Acid in Pediatric Patients with Epilepsy

Journal: Pharmaceuticals

Zhu et al. reported a study on the Association of ABCB1 Polymorphism with Therapeutic Response to Valproic Acid in Pediatric Patients with Epilepsy. The study is important as for as the efficacy and health risks of valproic acid are concerned in epileptic patients; however there are some issues in the study which need to be rectified before publication.  

1.     The rationale and novelties of the study must be properly described as plenty of work is reported on the topic. Discuss your work in the context of the reported works as below.

https://www.futuremedicine.com/doi/abs/10.2217/14622416.8.9.1151

https://www.frontiersin.org/articles/10.3389/fphar.2022.1037239/full

https://link.springer.com/article/10.1007/s10072-018-3681-y

2.     Mention the complete name on first mention such GG, and AG. Avoid to use the abbreviations in the abstract. 

3.     How the persistent seizure and seizure-free patients were differentiated?  It should be clarified in the introduction and methodology sections of the manuscript?

4.     The sentence is not clear and need to be rephrased; The patients included in this study were predominantly treated with VPA mono- 98 therapy (53.5%) as well as in the seizure-free gro…..

5.     If no significant differences in VPA concentration and AC were observed among all the participants then hoe the ADRs were different? Are these ADRs dose independent?

6.     To know about the real picture of the study all the pharmacokinetic parameters must be mentioned in the manuscript. 

7.     The correlation must be established in result section through graphical representation.

8.     The methodology section needs to be more descriptive to reproduce this work easily.

9.     The discussion section must be properly summarized and the outcomes of the study must be evaluated.

10.  What suggestions and strategies will be recommended to the health care providers in decision-making and holding effective interventions?   

11.  The manuscript must be revised as per journal approved format. The up to date references must be incorporated and grammatical and typographical mistakes must be rectified.  

Comments on the Quality of English Language

Moderate changes are required. 

Author Response

Dear reviewer,

Thank you very much for your careful review and constructive suggestions to our manuscript. We have addressed the comments and highlighted all the corrections in red in the revised manuscript. Point-by-point response to your comments and suggestions are listed below.

Comments 1: The rationale and novelties of the study must be properly described as plenty of work is reported on the topic. Discuss your work in the context of the reported works as below.

https://www.futuremedicine.com/doi/abs/10.2217/14622416.8.9.1151

https://www.frontiersin.org/articles/10.3389/fphar.2022.1037239/full

https://link.springer.com/article/10.1007/s10072-018-3681-y

Response 1: Thank you very much for your comments and we agree with you. In view of the extensive works that have already been described on this topic, we have added a discussion of the rationale and novelties of this study in the introduction (page 2, line 89-100 in the revised manuscript).

Comments 2: Mention the complete name on first mention such GG, and AG. Avoid to use the abbreviations in the abstract.

Response 2: Thanks very much for your comments. We have checked the abbreviations in the manuscript and used the complete name in the first reference. Abbreviations in the abstracts have been corrected to complete names.

Comments 3: How the persistent seizure and seizure-free patients were differentiated? It should be clarified in the introduction and methodology sections of the manuscript?

Response 3: Thank you very much for the comments raised by you. We have described the evaluation of drug response in the Materials and Methods section of the manuscript. As defined in previous studies (Br. J. Clin. Pharmacol. 2008; 66:304-307; Epilepsia. 2010; 51:1069-1077), patients were classified as persistent seizure when valproic acid monotherapy or therapy combined with other antiseizure medications for at least 12 months failed to achieve sustained seizure freedom at the maximal tolerated dose. Patients were considered seizure-free if they did not experience any type of seizure during valproic acid treatment for at least 1 year (page 12, line 393-397 in the revised manuscript).

Comments 4: The sentence is not clear and need to be rephrased; The patients included in this study were predominantly treated with VPA monotherapy (53.5%) as well as in the seizure-free gro…..

Response 4: Thank you very much for your valuable comments and we agree with you. We have made the following modification: the patients in the seizure-free group received VPA monotherapy more frequently (69.7%), in contrast to the persistent seizure group, where it was more common for patients to receive polytherapy (75.4%) (page 3, line 116-118 in the revised manuscript).

Comments 5: If no significant differences in VPA concentration and AC were observed among all the participants then hoe the ADRs were different? Are these ADRs dose independent?

Response 5: We sincerely thank your valuable comments and suggestions. VPA is associated with non-linear pharmacokinetics due to saturable plasma protein binding, resulting in large differences between individuals in the dose to plasma concentration relationship (Drug Monit. 2018; 40:526-548). Several reports failed to find a high correlation between the plasma VPA concentrations and the therapeutic or toxic effects of the drug (Pharmacogenet. Genomics. 2013; 23:236-241; PLoS One. 2015; 10:14). Physiological and pathological characteristics of patients such as age, gender, genetic factors, co-morbidities and co-medication can cause individual differences in response to medications. Therefore, we also did not find a high correlation between the plasma VPA concentrations and ADRs. We cannot currently determine whether these ADRs are dose independent, as some develop at low VPA doses while others develop at increased doses. Consideration should be given to exploring this correlation in larger sample populations.

Comments 6: To know about the real picture of the study all the pharmacokinetic parameters must be mentioned in the manuscript.

Response 6: Thank you very much for your valuable comments and we agree with you. We have complemented the description of the pharmacokinetic parameters of valproic acid in the introduction section of the manuscript to provide a comprehensive picture of this study (page 2, line 49-55 in the revised manuscript).

Comments 7: The correlation must be established in result section through graphical representation.

Response 7: Thank you very much for the comments raised by you. We have added graphs in the results section of the manuscript to show the association of ABCB1 polymorphism with the efficacy and adverse drug reactions of VPA.

Comments 8: The methodology section needs to be more descriptive to reproduce this work easily.

Response 8: Thank you very much for your insightful comments and we agree with you. We have given a more detailed description in the methodology section of the revised manuscript (page 12, line 404-406 and line 418-424 in the revised manuscript). The specific details can also be found in our previous research (Front. Neurosci. 2020; 14:14; Pediatr. Neurol. 2023; 146:55-64).

Comments 9: The discussion section must be properly summarized and the outcomes of the study must be evaluated.

Response 9: Thank you very much for the comments raised by you. We have marked the appropriate summary and assessment of the results of this study in the discussion section in red.

Comments 10: What suggestions and strategies will be recommended to the health care providers in decision-making and holding effective interventions? 

Response 10: Thanks very much for your comments. Recommendations for health providers as follows: Patients with AA genotype of ABCB1 gene rs1128503 may respond poorly to VPA therapy. For such patients, if seizure cannot be controlled after reaching the target dose, the combination or switch to other antiseizure medications should be considered. Digestive-related adverse drug reactions should be of concern in patients with AA or AG genotype of ABCB1 rs3789243, and cutaneous adverse drug reactions should be of concern in patients with GG genotype of ABCB1 rs3789243. For such patients, the dose can be increased slowly. If adverse drug reactions are not tolerated, switching to another antiseizure medication should be considered.

Comments 11: The manuscript must be revised as per journal approved format. The up to date references must be incorporated and grammatical and typographical mistakes must be rectified. 

Response 11: We sincerely thank your valuable comments and suggestions. We have carefully formatted the manuscript according to the requirements of the journal. Recent references in the manuscript are incorporated and grammatical mistakes have been checked.

Reviewer 3 Report

Comments and Suggestions for Authors

Line 97-The mean steady-state concentration of VPA was 62.68 ± 22.46

Authors should indicate it is a blood or serum concentration

Author Response

Dear reviewer,

Thank you very much for your careful review and constructive suggestions to our manuscript. We have addressed the comments and highlighted all the corrections in red in the revised manuscript. Point-by-point response to your comments and suggestions are listed below.

Comments 1: Line 97-The mean steady-state concentration of VPA was 62.68 ± 22.46. Authors should indicate it is a blood or serum concentration.

Response 1: Thank you very much for your comments and we agree with you. We have revised the description in the manuscript as "The mean steady-state serum concentration of VPA" (page 3, line 115 in the revised manuscript).

Round 2

Reviewer 2 Report

Comments and Suggestions for Authors

Minor grammatical and typogrphical errors must be rectified. 

Comments on the Quality of English Language

Minor grammatical changes must be carried out. 

Author Response

Thank you very much for your comments. We have carefully checked and corrected minor grammatical and typographical errors in the manuscript.